# Differential Effects of Optimism and Pessimism on Adolescents’ Subjective Well-Being: Mediating Roles of Reappraisal and Acceptance

**DOI:** 10.3390/ijerph19127067

**Published:** 2022-06-09

**Authors:** Rong Zou, Xiaobin Hong, Gaoxia Wei, Xia Xu, Jiajin Yuan

**Affiliations:** 1Hubei Key Laboratory of Sport Training and Monitoring, Department of Psychology, College of Health Science, Wuhan Sports University, Wuhan 430079, China; zourong4035@whsu.edu.cn (R.Z.); hongxiaobin1002@163.com (X.H.); 2CAS Key Laboratory of Mental Health, Institute of Psychology, Beijing 100101, China; weigx@psych.ac.cn; 3CAS Key Laboratory of Behavioral Science, Institute of Psychology, Beijing 100101, China; 4Department of Psychology, University of Chinese Academy of Sciences, Beijing 100101, China; 5Institute of Brain and Psychological Science, Sichuan Normal University, Chengdu 610066, China

**Keywords:** optimism, pessimism, reappraisal, acceptance, depression, life satisfaction, adolescent

## Abstract

Prior research has found the differential strength of optimism and pessimism in predicting physical health. However, whether similar findings would be obtained in predicting subjective well-being and the possible underlying mechanisms are still unclear. This study examined the relative strength of optimism and pessimism in predicting adolescent life satisfaction and depression, and further explored the possible mediating mechanisms from the perspective of emotion regulation. A sample of 2672 adolescents (*M*_age_ = 13.54 years, *SD* = 1.04; 55.60% boys) completed a survey assessing optimism and pessimism, the habitual use of reappraisal and acceptance strategies, life satisfaction, and depression. The results from dominance analysis revealed that the presence of optimism was more powerful than the absence of pessimism in predicting adolescent life satisfaction, while the absence of pessimism was more powerful than the presence of optimism in predicting adolescent depression. Moreover, mediation models showed that reappraisal and acceptance mediated both the link between optimism and life satisfaction and the link between pessimism and depression. These findings suggest possible avenues for intervening in different aspects of adolescent subjective well-being.

## 1. Introduction

The personality dimension dispositional optimism (expectancies regarding future outcomes) has been widely proven to show strong associations with markers of physical health and subjective well-being across ages and nations [1,2,3,4]. Especially in uncertain or stressful situations, such as the current COVID-19 pandemic, holding more positive and less negative expectancies about the future is important and protective for individual health [5,6]. Compared to pessimists, optimists have been linked to lower levels of markers of inflammation, lower cortisol responses under stress, more adaptive immune responses and post-traumatic growth, lower depression levels, and higher life satisfaction [7,8,9,10].

Most of the studies cited used the Life Orientation Test (LOT) [11] or its revision (LOT-R) developed by Scheier and his colleagues [12] to measure dispositional optimism, treating dispositional optimism as a unidimensional construct. Although there were some studies supporting the unidimensional view [13,14], more recently, mounting studies have suggested that optimism and pessimism seem to represent two independent constructs [4,15,16,17,18,19,20,21,22,23,24,25]. Evidence from behavior genetic studies has demonstrated that optimism and pessimism are distinct systems [15]. A review addressing the neural basis of optimism and pessimism has also indicated that optimism and pessimism are differentially associated with the two cerebral hemispheres [26]. Moreover, contrary to the unidimensional point of view, the developmental tendencies of optimism and pessimism among older people are not in opposite directions [22], which is further supported by the evidence from adolescent studies [23].

Tracing the theoretical root of dispositional optimism, optimism and pessimism are also related to different motivational processes [27]. The concepts of optimism and pessimism are linked to ideas of a long history of expectancy–value models of motivation. According to expectancy–value theories, behavior represents the pursuit of goals [27]. The higher the importance of a given goal to the individual, the greater its value. The other key concept of this motivational model is expectancy—that is, confidence that the goal can be reached. Optimism and pessimism are defined, respectively, by confidence or doubt in achieving the goal. This seems a simple difference between optimism and pessimism, but it also seems to matter a lot [8,27]. Anticipating generally good outcomes versus bad ones is related to motivational processes that influence individuals’ subsequent cognitive, affective, and volitional processes [27]. Positive expectations about the future are likely to be associated with the approach motivation system, causing individuals to be more sensitive to positive stimuli, to experience more positive emotions, to put more effort into their goals, and to be more likely to persevere even in the face of obstacles [27,28,29]. On the contrary, negative expectations about the future are likely to activate the avoidance motivation system, causing individuals to be more sensitive to negative stimuli, to experience more negative emotions such as anxiety and depression, and to withdraw effort prematurely [27,28,29]. The relative independence between the approach and the avoidance motivation systems [30], combined with the unique relationship between optimism and the approach motivation system, and that between pessimism and the avoidance motivation system, could further provide theoretical and indirect evidence for the bi-dimensional structure of optimism and pessimism.

Since optimism and pessimism are relatively independent, we may wonder whether the presence of optimism or the absence of pessimism is more prominent in predicting important adaptive outcomes. Answering this question could help us to specify the relative contributions of optimism and pessimism to the prediction of life outcomes, which may prove profitable both practically and theoretically [4]. However, most of the existing research on the benefits of dispositional optimism in various life outcomes struggles to answer this question, as they treat optimism and pessimism as bipolar in nature [8]. Occasionally, a handful of studies have separately explored the different roles of optimism and pessimism, mainly in the field of physical health [4,16,17,21,22]. Recently, a meta-analysis from Scheier et al. [4], based on data from 61 separate samples of previously published studies, separated optimism and pessimism and examined their relative strength in contributing to associations with health. Their results indicate that the absence of pessimism is a better predictor of physical health outcomes than the presence of optimism [4]. However, the outcomes examined regarding the differential strength of optimism and pessimism in most of the prior studies involved physical health [4,16,17,21,22]. It is uncertain whether comparable results would emerge if subjective well-being outcomes, such as life satisfaction and depression, were investigated.

Several previous studies explored the relative strength of optimism and pessimism in subjective well-being, but they mainly focused on adults and yielded inconsistent results [18,31,32]. So far, there have been only a few adolescent studies exploring the predictive effects of optimism and pessimism on subjective well-being outcomes separately [25,33]. Sulkers et al. found that optimism predicted positive aspects and pessimism predicted negative aspects of subjective well-being [33]. Tejada-Gallardo et al. found that only optimism but not pessimism was significantly related to happiness [25]. These studies provided valuable information for specifying the relative strength of optimism and pessimism in predicting adolescents’ subjective well-being. However, due to the special participants (adolescents with cancer) and the limited sample size (*N* = 33) [33], the findings may not be generalizable to healthy adolescents, who represent most of the adolescent population. Moreover, these studies were mainly based on correlation or regression coefficients to compare the relative strength of optimism and pessimism, resulting in less robust results [25,33]. Therefore, much further work is needed to clarify which is more important, optimism or pessimism, for adolescents’ subjective well-being. Exploring the differential impact of optimism and pessimism on subjective well-being among adolescents has important implications for intervention practices.

Adolescence is accompanied by various physiological, psychological, and social challenges, including increased potential for conflict with parents, more body image concerns, and an increase in the developmental tasks of identity formation and autonomy [34,35]. Adolescents entering middle school experience greater academic stress, facing the pressure of entrance examinations for high schools or colleges. Those changes expose adolescents to stressful situations frequently, leading to more frequent and intense negative emotions [35]. The immaturity of the prefrontal regions during adolescence, which is related to the ability to perform emotion regulation [36], may amplify adolescents’ vulnerability to stress, resulting in declined life satisfaction and heightened risk for psychopathology [37,38]. The onset of psychopathology in adolescence has a far-reaching impact and could predict mental health issues in adulthood [38,39]. Therefore, it is important to pay attention to adolescents’ subjective well-being and its possible antecedents.

As an important personality trait related to motivational processes [27], dispositional optimism exerts a great deal of influence on adolescents’ life course outcomes [3,27]. Positive expectations for the future are more likely to activate adolescents’ approach motivation system, allowing them to perceive and experience the world as if they are wearing rose-colored glasses [27,28]. Even if they are facing many changes and challenges, the approach motivation system may also cause adolescents to be more sensitive to positive stimuli and to experience more positive emotions [28,29]. As aforementioned, negative expectations about the future may activate adolescents’ avoidance motivation system [27,28]. This may cause adolescents to be more sensitive to negative stimuli. Meanwhile, it may also help them to cope with problems with the greater use of defensive and avoidant methods, which may aggravate their stress and negative emotions [40]. Therefore, expectations about the future are powerful inner forces that influence adolescents’ subjective well-being. As optimism and pessimism seem to have been differentially linked to positive and negative aspects of subjective well-being in previous studies [25,33], depression and life satisfaction, which are important components of subjective well-being, are chosen to represent the indicators of positive and negative aspects of subjective well-being, respectively, in this study. Based on the theoretical grounds of dispositional optimism and previous studies, we hypothesize that optimism would be a stronger predictor for adolescents’ life satisfaction than pessimism, and pessimism would be a stronger predictor for adolescents’ depression than optimism.

Although, according to our first hypothesis, optimism and pessimism would be related distinctively to adolescent subjective well-being, the underlying mechanisms of these associations are still unclear. The top-down model of personality, coping, and emotional outcomes [41] may lend us a useful framework for exploring this issue. The model posits that personality traits such as dispositional optimism can exert effects on the specific ways in which individuals regulate and respond to emotions, sequentially affecting adaptation outcomes [40,41]. Empirical research has supported this model by indicating that some personality traits, such as the Big Five personality traits or dispositional mindfulness, predicted subjective well-being or mental health through the mediating effects of emotion regulation (ER) [42,43]. Referring to dispositional optimism, optimistic adolescents have positive expectations for their future and believe that good outcomes require some effort [40]. Therefore, optimistic adolescents are more likely to use adaptive strategies to regulate negative emotions, sequentially promoting problem-solving and facilitating favorable outcomes when confronted with stressful situations or negative emotions [8,40,41]. Pessimistic adolescents, on the other hand, have negative expectations for the future and do not believe that positive outcomes will occur. For this reason, they are less likely to use adaptive strategies to regulate emotions in the face of pressure or negative emotions [40,41]. Accordingly, for pessimistic adolescents, stress and threats are likely to last and negative emotions may even be intensified, which is toxic for long-term subjective well-being.

Quite a few empirical studies speak directly to the relationship between dispositional optimism and ER. The available evidence consistently indicates that a higher level of optimism is linked to more habitual use of adaptive strategies to regulate emotions, such as reappraisal and acceptance, while a higher level of pessimism is linked to less habitual use of adaptive ER [44,45]. Reappraisal and acceptance are the two important and adaptive ER strategies with a high frequency of use and beneficial health outcomes [46,47,48]. Reappraisal, in many cases, is an effortful cognitive change strategy involving reframing the meaning of an emotional situation so that individuals feel better [49]. Acceptance is defined as a non-elaborative, non-evaluative and present-focused perspective, which allows individuals to accept thoughts, feelings, and sensations as they are [47,49]. Unlike reappraisal, acceptance does not involve the processes of reinterpretation, arguing, reasoning, inhibitory control, and response inhibition and, thus, requires less recruitment of cognitive and brain resources [47]. Therefore, in contrast with reappraisal, acceptance strategies may be much easier to implement successfully, especially for individuals with insufficient cognitive resources. Pessimistic adolescents’ cognitive resources are likely to be overwhelmed by negative thoughts and emotions [27,35,36]. The use of the acceptance strategy that requires fewer cognitive resources may have special significance for pessimistic adolescents [47]. However, the use of both reappraisal and acceptance has been proven effective in the protection of subjective well-being [47,50]. The use of reappraisal and acceptance cannot only immediately reduce adolescents’ negative emotions [51] but may also be associated with less depression [46,52] and higher life satisfaction in the long term [53]. Based on the top-down model of personality, coping and emotional outcomes [41], and previous studies, we hypothesize that reappraisal and acceptance would mediate the relationship between dispositional optimism and subjective well-being.

To summarize, this study aimed to explore the differences in the predictive power of optimism and pessimism on positive and negative aspects of adolescents’ subjective well-being (i.e., life satisfaction and depression), as well as the possible mediating roles of reappraisal and acceptance. Based on the literature review, the present study tested the following three hypotheses: (1) optimism would be a stronger predictor for adolescents’ life satisfaction than pessimism, and pessimism would be a stronger predictor for adolescents’ depression than optimism; (2) reappraisal and acceptance would mediate the relationship between optimism and adolescents’ life satisfaction; and (3) reappraisal and acceptance would mediate the relationship between pessimism and adolescents’ depression.

## 2. Materials and Methods

### 2.1. Participants and Procedure

Participants included 2672 adolescents (age range: 11.09–17.76 years old, *M*_age_ = 13.54 years, *SD* = 1.04, 1486 boys (55.60%)) recruited from two middle schools in central and southwest China. No participants reported any records of using psychiatric medication. There was less than 1% missing data, and the missing data were estimated with the Expectation Maximization (EM) procedure in SPSS 25.0 (SPSS, Chicago, IL, USA).

We first submitted our research plan to the university’s Ethical Committee for Scientific Research and received approval. Then, the survey was conducted in classrooms after informed consent was obtained from the class teachers and participants’ parents. Trained graduate students of psychology explained the requirements of the survey using standard instructions emphasizing the authenticity, independence, and integrity of all answers.

### 2.2. Measures

#### 2.2.1. Optimism and Pessimism Scale

Optimism and pessimism were assessed by the Chinese version of the Life Orientation Test-Revised (CLOT-R) [54], which consists of ten active items and two filler items (e.g., It’s important for me to keep busy). Five positively worded items (e.g., When things are bad, I expect them to go better) constitute the optimism subscale, and five negatively worded items (e.g., I hardly ever expect things to go my way) constitute the pessimism subscale. Participants indicated the extent to which they agreed with each statement on a 5-point scale ranging from 1 (strongly disagree) to 5 (strongly agree). The CLOT-R has been used in Chinese adolescents with good reliability and validity [23,55]. Its bi-dimensional structure among Chinese adolescents was also supported in previous studies [23,54]. In this study, the index of optimism–pessimism bi-dimensional confirmatory factor analysis (CFA) showed a good fit: χ^2^/df = 12.02, RMSEA = 0.06, CFI = 0.95, TLI = 0.94, SRMR = 0.03, significantly better than the one-factor solution: χ^2^/df = 36.00, RMSEA = 0.11, CFI = 0.84, TLI = 0.80, SRMR = 0.07. Optimism was negatively correlated with pessimism in this study (*r* = −0. 51, *p* < 0.001). Combined with the results of CFA and other previous studies [4,24,25], it showed that optimism and pessimism were related but distinct concepts. The Cronbach’s α coefficient and the McDonald’s omega for the optimism subscale were 0.74 and 0.74, respectively, while both the Cronbach’s α and the McDonald’s omega for the pessimism subscale were 0.82.

#### 2.2.2. Reappraisal

Reappraisal is measured by six items (e.g., When I’m faced with a stressful situation, I make myself think about it in a way that helps me stay calm) selected from the Emotion Regulation Questionnaire (ERQ) [56], which is widely used in Chinese adolescents with sufficient internal consistency and validity [57]. Items were rated on a 7-point Likert scale ranging from 1 (strongly disagree) to 7 (strongly agree). Higher scores indicated more usage of reappraisal in regulating emotions. The Cronbach’s α coefficient and the McDonald’s omega in this sample were 0.84 and 0.84, respectively.

#### 2.2.3. Acceptance

Acceptance was assessed by the Chinese version of Acceptance and Action Questionnaire II (AAQ-II), which has been used among Chinese adolescents with good internal consistency and validity [58]. It included seven items (e.g., I’m afraid of my feelings) rated on a 7-point Likert scale, ranging from 1 (never) to 7 (always). Reversed scores of all seven items were summed up to form the total scores, with higher total scores reflecting a higher tendency to use the acceptance strategy. The Cronbach’s α coefficient and the McDonald’s omega of the scale in this study were 0.92 and 0.92, respectively.

#### 2.2.4. Subjective Well-Being

Adolescents’ subjective well-being in this study was indicated by life satisfaction and depression. Life satisfaction was assessed by the Satisfaction with Life Scale (SWLS) [59]. It included five items (e.g., I am satisfied with my life) rated on a 7-point Likert scale, ranging from 1 (strongly disagree) to 7 (strongly agree). A high score represented high life satisfaction. SWLS has good internal consistency and validity when measuring life satisfaction in Chinese children and adolescents [9,55]. The Cronbach’s α coefficient and the McDonald’s omega in this study were 0.78 and 0.78, respectively.

Adolescents’ depression was measured by the Chinese version of the Center for Epidemiologic Studies Depression Scale (CES-D) [60], which is well validated among Chinese adolescents [61]. It consists of 20 items, with 16 items reflecting negative symptoms (e.g., I felt depressed) and four reverse-coded items reflecting positive states (e.g., I am happy). Participants were asked to report the frequency of events and ideas over the past week on a 4-point Likert scale ranging from 0 (rarely or none of the time) to 3 (most or all of the time). Higher total scores of this scale indicated more severe depressive symptoms. The Cronbach’s alpha coefficient and the McDonald’s omega in this study were 0.80 and 0.81, respectively.

#### 2.2.5. Demographic Information

Besides the above scales, adolescents also completed a questionnaire soliciting information about sex, age, and family socioeconomic status (SES). Family SES was assessed by the Family Affluence Scale (FAS), which is well established with moderate internal consistency and good validity and widely used among adolescents both in China and in Western countries [62,63]. FAS includes four items indicating family affluence (e.g., Do you have your own bedroom for yourself?). High total scores represent high affluence. In line with previous studies [9,23,62,63], the Cronbach’s α coefficient and the McDonald’s omega of FAS in this sample were 0.60 and 0.67, respectively.

### 2.3. Statistical Analysis

To address the study’s first aim (determining the relative contributions of optimism and pessimism to explaining variances in adolescents’ life satisfaction and depression), we conducted dominance analyses to identify the relative importance of predictors in multiple regression based on an examination of the *R*^2^ values for all possible subset models [64,65].

To address the study’s second aim (exploring how adolescents’ optimism or pessimism predict their life satisfaction and depression), mediation models were tested by PROCESS macro for SPSS (Version 3, Model 4) [66], which has been widely used by many scholars to test mediation models [57,67]. PROCESS calculates standardized direct or indirect effects using bootstrapping analyses with 10,000 replications. If the bias-corrected 95% confidence interval does not contain zero, it indicates that the effect is significant. To yield standardized coefficients, all variables (excluding sex) were converted to z-scores prior to mediation analysis.

To avoid common method bias, we performed Harman’s single-factor test on all items of the current study before data analysis [68]. The results show that there were 13 factors whose eigenvalues were greater than 1. These factors totally accounted for 56.77% of variances of all variables. The first single factor only accounted for 26.88% of variance, which is less than the critical value of 40%, arguing against the presence of significant measurement errors such as common method bias.

## 3. Results

### 3.1. Preliminary Analyses

Descriptive statistics and correlations among variables in this study are displayed in Table 1. Optimism was positively correlated with reappraisal, acceptance, and life satisfaction and negatively correlated with depression. Pessimism was negatively correlated with reappraisal, acceptance, and life satisfaction and positively correlated with depression. Moreover, both reappraisal and acceptance were positively associated with life satisfaction and negatively related with depression.

As Table 1 demonstrates, there were some correlations with demographic variables. Both age and SES were significantly related to life satisfaction, and SES and sex were significantly correlated with depression. Following the principles of selecting control variables [69], in subsequent analyses, age and SES were set as covariates when the outcome variable was life satisfaction, and age, SES, and sex were also set as control variables when the outcome variable was depression.

### 3.2. Dominance Analysis

In this stage, we performed a series of hierarchical regression analyses, with control variables entered in the first step and optimism or pessimism of interest in the second or third step (method = enter). As shown in Table 2, the results of the dominance analysis using life satisfaction as the outcome variable established complete dominance for optimism relative to pessimism, as the additional contribution of optimism in predicting life satisfaction was higher in all subset models relative to the additional contribution of pessimism (*k* = 0 (indicating that there were only control variables in the regression model), 0.214 > 0.093; *k* = 1 (indicating that the regression model included a predictor in addition to control variables), 0.131 > 0.010). Consequently, optimism completely dominated pessimism in predicting adolescents’ life satisfaction.

As seen in Table 2, when using depression as the outcome variable, complete dominance for pessimism relative to optimism was established, as the additional contribution of pessimism in predicting depression was higher in all the subset models relative to the additional contribution of optimism (*k* = 0 (indicating that there were only control variables in the regression model), 0.296 > 0.246; *k* = 1 (indicating that the regression model included a predictor in addition to control variables), 0.125 > 0.074). Consequently, pessimism completely dominated optimism in predicting adolescents’ depression.

### 3.3. Mediation Models Test

The results of dominance analysis show that optimism completely dominated pessimism in predicting adolescents’ life satisfaction, while pessimism completely dominated optimism in predicting adolescents’ depression. In this stage, through testing mediation models with reappraisal and acceptance as mediators, we further explored how optimism predicted life satisfaction when controlling pessimism and how pessimism predicted depression when controlling optimism.

In Table 3 and Figure 1a, which contain the findings of multiple mediation model analyses to predict life satisfaction from optimism, reappraisal and acceptance are presented. The regression coefficient of reappraisal predicted by optimism was 0.33 (95% CI [0.36, 0.43]) and that of life satisfaction predicted by reappraisal was 0.16 (95% CI [0.13, 0.20]), producing a significant indirect effect of 0.05 (95% CI [0.04, 0.07]). The regression coefficient of acceptance predicted by optimism was 0.20 (95% CI [0.17, 0.24]) and that of life satisfaction predicted by acceptance was 0.18 (95% CI [0.15, 0.22]), indicating a significant indirect effect of 0.04 (95% CI [0.03, 0.05]). The total indirect effect with both mediators was 0.09, 95% CI [0.07, 0.11], accounting for 20.93% of the total effect of optimism on life satisfaction. Both reappraisal and acceptance showed significant indirect effects on the relationship between optimism and life satisfaction, although there was no significant difference in indirect effects between the two mediating paths (effect difference of paths = 0.02, 95% CI [−0.003, 0.04].

The findings of multiple mediation model analysis to predict depression from pessimism, reappraisal, and acceptance are presented in Table 4. The regression coefficient of reappraisal predicted by pessimism was −0.13 (95% CI [−0.17, −0.09]) and that of depression predicted by reappraisal was −0.10 (95% CI [−0.12, −0.07]), producing a significant indirect effect of 0.01 (95% CI [0.006, 0.02]). The regression coefficient of acceptance predicted by pessimism was −0.42 (95% CI [−0.46, −0.39]) and that of depression predicted by acceptance was −0.50 (95% CI [−0.53, −0.47]), indicating a significant indirect effect of 0.21 (95% CI [0.18, 0.24]). The total indirect effect with both mediators was 0.22, 95% CI [0.20, 0.25], accounting for 53.66% of the total effect of pessimism on depression. Both reappraisal and acceptance showed significant indirect effects on the relationship between pessimism and depression. Moreover, the difference in indirect effects between the mediating paths was significant (effect difference of paths = −0.20, 95% CI [−0.23, −0.17]). Specifically, the mediating role of acceptance in the relationship between pessimism and depression is greater than that of reappraisal. The verified mediation model is shown in Figure 1b.

## 4. Discussion

The current study, using a large sample of normal adolescents (*N* = 2672) and robust dominance analysis, examined the relative strength of optimism and pessimism in contributing to associations with adolescents’ subjective well-being (i.e., life satisfaction and depression). It further explored the mediating roles of reappraisal and acceptance in these associations based on the top-down model of personality, coping, and emotional outcomes [41]. Many previous studies have explored the relationship between dispositional optimism and adolescents’ subjective well-being in different cultures or nations [3], but most of them treated dispositional optimism as bipolar, which made it impossible to compare the relative strength of optimism and pessimism in predicting adolescents’ well-being. The current study treated optimism and pessimism as two related but separate constructs. To our knowledge, the present study is one of the first studies to date to specify the relative contributions of optimism and pessimism to the prediction of life satisfaction and depression and further explore the possible mediating mechanisms underlying these associations in Chinese adolescents. Examining the differential effects of optimism and pessimism teases apart the nuances of them in predicting the positive and negative aspects of subjective well-being in adolescents. It extends the domain from physical health to subjective well-being in which optimism and pessimism show differential strength. Combining the findings about the underlying mediating mechanisms, this study also provides insights into how the presence of optimism or the absence of pessimism relates to adolescents’ life satisfaction and depression from the perspective of emotion regulation. We discuss each of the research questions based on the results of dominance analysis and mediation models tests.

First, consistent with our hypothesis, we found that the presence of optimism was more powerful than the absence of pessimism in predicting adolescent life satisfaction, while the absence of pessimism was more powerful than the presence of optimism in predicting adolescent depression. These findings are congruent with the theoretical grounding that optimism and pessimism are related to different motivational processes and may show differential effects on life outcomes [8,27,28], as well as the previous study in adolescents [25,33]. By expanding on previous research [18,31,32,33], we not only use a large adolescent sample but also adopt the robust dominance analysis rather than just the size comparison of correlation or regression coefficients to examine the relative strength of optimism and pessimism, which guarantees more reliable results. Our findings imply that, even though adolescence is a period of storm and stress accompanied by various changes, positive expectations about the future could motivate youth to perceive, appraise, and react to those changes and stress positively [28,29], indicating the special boosting effects of the presence of optimism on positive aspects of adolescents’ subjective well-being. Meanwhile, anticipating bad outcomes in the future may aggravate adolescents’ vulnerability to various changes and stress, causing them to be more susceptible to negative stimuli [29] and causing them to appraise and react to changes and stress in negative, avoidant, and defensive ways [40]. The stronger strength of pessimism in depression outlines the special protecting effects of the absence of pessimism on negative aspect of adolescents’ subjective well-being. These contrasting findings may reflect the differential influence of approach and avoidance motivation systems relating to optimism and pessimism, respectively. In the physical health domain, the absence of pessimism was consistently shown to be a better predictor compared to the presence of optimism [4,22]. However, in the domain of subjective well-being, the presence of optimism and the absence of pessimism show differential significance in positive and negative aspects. This echoes the statement of Scheier et al. [4] that it is important not to extrapolate the findings obtained within the physical health domain to possible findings involving other domains. The novel findings underline the importance of exploring the differential impact of optimism and pessimism on the domain of subjective well-being. These results also provide a more detailed reference for intervening in different aspects of subjective well-being among adolescents.

Second, consistent with the top-down model of personality, coping, and emotional outcomes [41] as well as our hypotheses, reappraisal and acceptance mediated both the association between optimism and adolescent life satisfaction and the association between pessimism and adolescent depression. In line with previous studies, our results show that the presence of optimism or the absence of pessimism was associated with the more habitual use of reappraisal and acceptance strategies [44,45], which, in turn, was associated with greater life satisfaction and less depression [46,52,53]. Adolescence is a period of storm and stress, as well as a period of heightened risk for psychopathology [38]. The results of the mediation model in this study not only address the importance of dispositional optimism to adolescents’ subjective well-being but also answer the key question of how dispositional optimism predicts adolescents’ subjective well-being from the perspective of emotion regulation, providing path reference and insights into targeted interventions for promoting adolescents’ subjective well-being.

Another interesting result in this study is that the mediating effect of acceptance was significantly greater than that of reappraisal in the relationship between pessimism and adolescent depression, while there was no significant difference in the mediating effects of acceptance and reappraisal in the relationship between optimism and adolescent life satisfaction. This may be related to the characteristics of reappraisal and acceptance strategies and the cognitive characteristics of optimistic and pessimistic adolescents. For adolescents, the ability to regulate emotions is a pivotal skill for both their physical health and subjective well-being [70]. There are various strategies for regulating emotions, and they differ in effectiveness and implementation effort. The use of both reappraisal and acceptance has been proven to be effective in reducing negative emotions and adaptive for subjective well-being in the long term [46,47,48]. However, compared to implementing a reappraisal strategy, the use of an acceptance strategy requires less recruitment of cognitive and brain resources [47]. Meanwhile, compared with optimists, pessimists have negative expectations about the future, so their limited cognitive resources are more likely to be occupied by negative thoughts, leaving insufficient cognitive resources for emotion regulation [35,36,71]. Therefore, in contrast with reappraisal, acceptance strategies may be more likely to be implemented successfully and play a positive role in reducing depression, especially for pessimistic adolescents with insufficient cognitive resources. For optimistic adolescents, they hold positive expectancies for their future. They pay more attention to positive information and have more cognitive capacity for emotion regulation compared with pessimistic adolescents [29,72]. This may lead to the result that reappraisal and acceptance play an equally important mediating role in the association between optimism and life satisfaction. This finding highlights the special significance of the acceptance strategy for pessimistic adolescents. It could provide a reference for choosing which adaptive emotion regulation strategies to use when intervening in the development of depression in pessimistic adolescents.

The current study has several limitations that should be highlighted. First, the cross-sectional design of this exploratory study can provide valuable information for the research questions, but we should bear in mind that the findings are based on correlational data, which cannot support causal inferences. Future longitudinal or experimental studies may be needed to clarify causal directions. Second, life satisfaction and depression were chosen as indicators of positive and negative aspects of subjective well-being, which cannot cover all the aspects of subjective well-being. Future studies could test the relative contributions of optimism and pessimism to the prediction of other important adaptive indicators, providing the basis for targeted interventions. Third, the variable-centered approach used in this study cannot allow for exploring how optimism and pessimism, two separate but related traits, are integrated within an individual and what the adaptive outcomes of individuals with different combinations of optimism–pessimism levels are. Blasco-Belled et al. [24] adopted the person-centered approach to identify different types of individuals based on the combinations of optimism and pessimism and their relationship to emotional intelligence, happiness, and life satisfaction. Future studies could combine the variable-centered and person-centered approaches to explore the relationship between personality and adolescent subjective well-being comprehensively. Last, we only examined the possible mediating effects of two adaptive emotion regulation strategies, reappraisal and acceptance, in the associations between dispositional optimism and adolescents’ subjective well-being, which was unable to analyze which emotion regulation strategies were most suitable for optimistic or pessimistic adolescents. Although reappraisal and acceptance are two of the most frequently used emotion regulation strategies that play an important role in health according to existing research results [46,47,48], adolescents may use various emotion regulation strategies in real life [46]. In the future, longitudinal behavioral and neuroimaging studies could be used to systematically compare the efficiency of implementing different emotion regulation strategies in optimistic or pessimistic adolescents, which will provide a key basis for interventions that could offer the most gain for the subjective well-being of optimistic and pessimistic adolescents.

## 5. Conclusions

Our study shows that the presence of optimism is more powerful than the absence of pessimism in predicting adolescent life satisfaction, while the absence of pessimism is more powerful than the presence of optimism in predicting adolescent depression. Moreover, reappraisal and acceptance mediated both the link between optimism and life satisfaction and the link between pessimism and depression. The mediating effect of acceptance is significantly greater than that of reappraisal in the relationship between pessimism and adolescent depression. These findings tease apart the nuances of optimism and pessimism in predicting adolescent life satisfaction and depression and provide insights into how to intervene for different aspects of adolescents’ subjective well-being.

## Figures and Tables

**Figure 1 ijerph-19-07067-f001:**
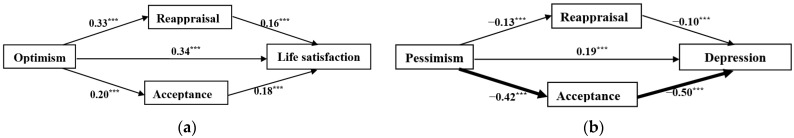
The verified mediation models in this study (*N* = 2672). (**a**) depicts standardized pathways from optimism to life satisfaction via reappraisal and acceptance. There was no significant difference in the mediating effect size between the two mediating paths. (**b**) depicts standardized pathways from pessimism to depression via reappraisal and acceptance. The mediating effect of acceptance was significantly greater than that of reappraisal in the relationship between pessimism and depression. Thickened path lines represent more prominent mediating paths. *** *p* < 0.001.

**Table 1 ijerph-19-07067-t001:** Descriptive statistics and inter-correlations between variables (*N* = 2672).

Variables	1	2	3	4	5	6	7	8	9
1. Age	-								
2. Sex	−0.02	-							
3. SES	0.13 ***	0.00	-						
4. OP	0.05 *	0.08 ***	0.28 ***	-					
5. PE	0.05 **	0.06 **	−0.25 ***	−0.51 ***	-				
6. RE	0.00	−0.05 *	0.19 ***	0.42 ***	−0.31 ***	-			
7. AC	−0.15 ***	−0.09 ***	0.27 ***	0.45 ***	−0.55 ***	0.25 ***	-		
8. LS	−0.15 ***	−0.03	0.28 ***	0.52 ***	−0.37 ***	0.37 ***	0.42 ***	-	
9. DE	0.07 ***	0.09 ***	−0.25 ***	−0.55 ***	0.59 ***	−0.36 ***	−0.71 ***	−0.46 ***	-
Mean	13.54	0.44	5.37	17.21	13.63	28.82	34.86	20.63	36.06
SD	1.04	0.50	2.34	4.18	4.62	7.01	10.02	6.36	11.48
Minimum	11.09	0	0	5	5	6	7	5	20
Maximum	17.76	1	9	25	25	42	49	35	79

*Note*. M = Mean, SD = standard deviation, SES = socioeconomic status, OP = optimism, PE = pessimism, RE = reappraisal, AC = acceptance, LS = life satisfaction, DE = depression. Sex was dummy coded such that boys = 0, girls = 1. * *p* < 0.05, ** *p* < 0.01, **** p* < 0.001.

**Table 2 ijerph-19-07067-t002:** Dominance analyses of predictors of life satisfaction and depression (*N* = 2672).

			Additional Contribution of	
Outcome Variables	Subset Model	R2	OP	PE
Life satisfaction	*k* = 0 average	0.091	0.214	0.093
	OP	0.305	-	0.010
	PE	0.183	0.131	-
	*k* = 1 average		0.131	0.010
	OP, PE	0.314	-	-
	Overall average		0.172	0.051
Depression	*k* = 0 average	0.070	0.246	0.296
	OP	0.315	-	0.125
	PE	0.366	0.074	-
	*k* = 1 average		0.074	0.125
	OP, PE	0.440	-	-
	Overall average		0.160	0.210

*Note*. OP = optimism, PE = pessimism; *k* = number of predictors (optimism or pessimism) besides covariates in multiple regression model.

**Table 3 ijerph-19-07067-t003:** Results of mediation models to predict life satisfaction from optimism, reappraisal, and acceptance.

Outcome	Predictors	*R* ^2^	*F*	*Estimate*	*SE*	95% CI
RE	OP	0.19	159.14 ***	0.33	0.02	**[0.36, 0.43]**
	Age			0.03	0.02	[−0.01, 0.06]
	SES			0.08	0.02	**[0.05, 0.13]**
	PE			−0.13	0.02	**[−0.17, −0.09]**
AC	OP	0.37	384.42 ***	0.20	0.02	**[0.17, 0.24]**
	Age			−0.10	0.02	**[−0.13, −0.07]**
	SES			0.10	0.02	**[0.06, 0.13]**
	PE			−0.42	0.02	**[−0.45, −0.39]**
LS	OP	0.36	247.10 ***	0.34	0.02	**[0.30, 0.38]**
	RE			0.16	0.02	**[0.13, 0.20]**
	AC			0.18	0.02	**[0.15, 0.22]**
	Age			−0.09	0.02	**[−0.12, −0.06]**
	SES			0.09	0.02	**[0.06, 0.13]**
	PE			−0.02	0.02	[−0.06, 0.02]
Indirect Effects						
P1: OP → RE → LS				0.05	0.01	**[0.04, 0.07]**
P2: OP → AC → LS				0.04	0.01	**[0.03, 0.05]**
Total				0.09	0.01	**[0.07, 0.11]**
Difference of the paths (P1–P2)				0.02	0.01	[−0.003, 0.04]

*Note*. *N* = 2672. All estimate values were standardized betas. SES = socioeconomic status, OP = optimism, PE = pessimism, RE = reappraisal, AC = acceptance, LS = life satisfaction, CI = confidence interval. Bolded confidence intervals do not include a zero, indicating a significant effect. *** *p* < 0.001.

**Table 4 ijerph-19-07067-t004:** Results of mediation models to predict depression from pessimism, reappraisal, and acceptance.

Outcome	Predictors	*R^2^*	*F*	*Estimate*	*SE*	95% CI
RE	PE	0.19	159.14 ***	−0.13	0.02	**[** **−0.17,** **−0.09]**
	Age			0.03	0.02	[−0.003, 0.07]
	SES			0.08	0.02	**[0.04, 0.11]**
	SEX			−0.03	0.04	[−0.09, 0.04]
	OP			0.33	0.02	**[0.29, 0.37]**
AC	PE	0.37	384.42 ***	−0.42	0.02	**[** **−0.46,** **−0.39]**
	Age			−0.10	0.02	**[** **−0.13,** **−0.07]**
	SES			0.10	0.02	**[0.06, 0.13]**
	SEX			0.09	0.03	**[0.03, 0.16]**
	OP			0.20	0.02	**[0.17, 0.24]**
DE	PE	0.61	685.24 ***	0.19	0.02	**[0.16, 0.22]**
	RE			−0.10	0.02	**[** **−0.12,** **−0.07]**
	AC			−0.50	0.02	**[** **−0.53,** **−0.47]**
	Age			−0.02	0.02	[−0.05, 0.001]
	SES			0.003	0.02	[−0.02, 0.03]
	SEX			0.03	0.02	[−0.02, 0.08]
	OP			−0.19	0.02	**[** **−0.22,** **−0.16]**
Indirect Effects						
P1: PE → RE → DE				0.01	0.003	**[0.006, 0.02]**
P2: PE → AC → DE				0.21	0.01	**[0.18, 0.24]**
Total				0.22	0.01	**[0.20, 0.25]**
Difference of the paths (P1–P2)				−0.20	0.01	**[** **−0.23,** **−0.17]**

*Note*. *N* = 2672. All estimate values were standardized betas. SES = socioeconomic status, OP = optimism, PE = pessimism, RE = reappraisal, AC = acceptance, DE = depression, CI = confidence interval. Bolded confidence intervals do not include a zero, indicating a significant effect. *** *p* < 0.001.

## Data Availability

Data are available, upon reasonable request, by emailing: yuanjiajin168@126.com or yuanjiajin168@sicnu.edu.cn.

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
