# Peer review of "Differential Effects of Optimism and Pessimism on Adolescents’ Subjective Well-Being: Mediating Roles of Reappraisal and Acceptance"

_ijerph, 2022, doi:10.3390/ijerph19127067_

Round 1

Reviewer 1 Report

The submitted manuscript seems pertinent to me, for its publication in this journal, however I am going to make some contributions or suggestions to improve the article.

The title, abstract and keywords are accurate.

The introduction contemplates a review of the state of the issue at the international level, in fact 31 and of the 70 citations are from the last 5 years, so the thematic current index is high, close to 50% for the last 5 years.

However, there are some works in prestigious journals based on the study of optimism-pessimism and life satisfaction, which should be cited, as they treat and address the same variables and are carried out in similar contexts but from other countries,

The participants in the study are 2,672 adolescents, so it can be considered a large sample size. The instruments used are relevant and the reliability of the scores obtained through the alpha and omega coefficients is adequately reported.

The results are based on tests that are relevant (relationship analysis, Dominance analysis and Mediation models test).

But the authors should include in their study:

- A broader discussion comparing the results with more international studies that address the same variables under study.

- In the conclusions they must contemplate and delve into the limitations found in their study and what the future lines of action will be in the future.

Author Response

Response to Reviewer 1 Comments

Point 1: The submitted manuscript seems pertinent to me, for its publication in this journal, however I am going to make some contributions or suggestions to improve the article.

Response 1: We would like to express our sincere thanks to you for the constructive and positive comments.

Point 2: The title, abstract and keywords are accurate.

Response 2: Thanks a lot for your comments. According to the suggestion from the Reviewer 3, we check the differences and similarities about the connotation between psychological well-being and subjective well-being. As the Reviewer 3 suggested, psychological well-being focuses on meaning in life, potential and self-realization, while subjective well-being focuses on individual enjoyment (Peng & Chen, 2010). What the present study targets are life satisfaction and depression that are in line with the connotation of subjective well-being. So the concept psychological well-being is replaced by subjective well-being in the revised manuscript.

Peng, Y.; Chen, H. Reconstruction on the connotation of well-being based on an integrative perspective. Adv. Psychol. Sci. 2010, 18, 1052-1061. (in Chinese)

Point 3: The introduction contemplates a review of the state of the issue at the international level, in fact 31 and of the 70 citations are from the last 5 years, so the thematic current index is high, close to 50% for the last 5 years.

However, there are some works in prestigious journals based on the study of optimism-pessimism and life satisfaction, which should be cited, as they treat and address the same variables and are carried out in similar contexts but from other countries;

Response 3: Thanks for your constructive suggestion. According to your kind suggestion, we searched relevant literature, cited these studies most closely related to this research topic in the part of Introduction, and compared these findings in the the part of Discussion. The amendments are marked in the revised manuscript. The added literature is mainly as follows:

  1. Blasco-Belled, A.; Rogoza, R.; Torrelles-Nadal, C.; Alsinet, C. Differentiating optimists from pessimists in the prediction of emotional intelligence, happiness, and life satisfaction: A latent profile analysis. Happiness Stud. 2022, 1-17. https://doi.org/10.1007/s10902-022-00507-4
  2. Tejada-Gallardo, C.; Blasco-Belled, A.; Torrelles-Nadal, C.; Alsinet, C. How does emotional intelligence predict happiness, optimism, and pessimism in adolescence? Investigating the relationship from the bifactor model. Psychol. 2020, 1-11. https://doi.org/10.1007/s12144-020-01061-z
  3. Chang, E. C.; Maydeu-Olivares, A.; D'Zurilla, T. J. Optimism and pessimism as partially independent constructs: Relationship to positive and negative affectivity and psychological well-being. Indiv. Differ. 1997, 23, 433-440. https://doi.org/10.1016/S0191-8869(97)80009-8

Point 4: The participants in the study are 2,672 adolescents, so it can be considered a large sample size. The instruments used are relevant and the reliability of the scores obtained through the alpha and omega coefficients is adequately reported.

The results are based on tests that are relevant (relationship analysis, Dominance analysis and Mediation models test).

Response 4: Thanks for your positive comments.

Point 5: But the authors should include in their study:

- A broader discussion comparing the results with more international studies that address the same variables under study.

Response 5: Thanks a lot for your kind suggestions. According to your suggestions, we searched and cited the research most relevant to the present study, and compared the results with these cited studies that address the same variables. The rewritten paragraph comparing the results in the Discussion is as follows:

First, consistent with our hypothesis, we found that the presence of optimism was more powerful than the absence of pessimism in predicting adolescent life satisfaction, while the absence of pessimism was more powerful than the presence of optimism in predicting adolescent depression. These findings are congruent with the theoretical grounding that optimism and pessimism are related to different motivational processes and may show differential effects on life outcomes [8,27,28], as well as the previous study in adolescents [25,33]. By expanding on previous research [18,31,32,33], we not only use a large adolescent sample, but also adopt the robust dominance analysis rather than just the size comparison of correlation or regression coefficients to examine the relative strength of optimism and pessimism, which guarantees more reliable results. Our findings imply that even though adolescence is a period of storm and stress accompanied by various changes, positive expectations about the future could moti-vate youth to perceive, appraise and react to those changes and stress positively [28,29], indicating the special boosting effects of the presence of optimism on positive aspect of adolescents’ subjective well-being. Meanwhile, anticipating bad outcomes in the future may aggravate adolescents’ vulnerability to various changes and stress, causing them to be more susceptible to negative stimuli [29], and causing them to appraise and react to changes and stress in negative, avoidant and defensive way [40]. The stronger strength of pessimism in depression outlines the special protecting effects of the absence of pessimism on negative aspect of adolescents’ subjective well-being. These contrasting findings may reflect the differential influence of approach and avoidance motivation systems relating to optimism and pessimism, respectively. In the physical health domain, the absence of pessimism was consistently shown to be a better predictor compared to the presence of optimism [4, 22]. However, in the domain of subjective well-being, the presence of optimism and the absence of pessimism show differential significance in positive and negative aspects. This echoes the statement of Scheier et al. [4] that it is important not to extrapolate the findings obtained within the physical health domain to possible findings involving other domains. The novel findings underline the importance of exploring the differential impact of optimism and pessimism on the domain of subjective well-being. These results also provide a more detailed reference for intervening on different aspects of subjective well-being among adolescents.

Point 6: In the conclusions they must contemplate and delve into the limitations found in their study and what the future lines of action will be in the future.

Response 6 : Thanks a lot for your constructive suggestions. According to your suggestions, we contemplated and delved into the limitations, and further proposed the future directions based on the limitations found in the present study. The rewritten paragraph about the limitations and the future directions is as follows:

The current study has several limitations that should be highlighted. First, the cross-sectional design of this exploratory study can provide valuable information for the research questions, but we should bear in mind that the findings are based on correlational data, which cannot support causal inferences. Future longitudinal or experimental studies may be needed to clarify causal directions. Second, life satisfaction and depression were chosen as indicators of positive and negative aspects of subjective well-being, which cannot cover all the aspects of subjective well-being. Future studies could test the relative contributions of optimism and pessimism to the prediction of other important adaptive indicators, providing the basis for targeted interventions. Third, the variable-centered approach used in this study cannot allow for exploring how optimism and pessimism, two separate but related traits, are integrated within an individual, and what the adaptive outcomes of individuals with different combinations of optimism-pessimism levels are. Blasco-Belled et al. [24] adopted the person-centered approach to identify different types of individuals based on the combinations of optimism and pessimism and their relationship to emotional intelligence, happiness, and life satisfaction. Future studies could combine the variable-centered and person-centered approaches to explore the relationship between personality and adolescent subjective well-being comprehensively. Last, we only examined the possible mediating effects of two adaptive emotion regulation strategies, reappraisal and acceptance, in the associations between dispositional optimism and adolescents' subjective well-being, which was unable to analyze which emotion regulation strategies were most suitable for optimistic or pessimistic adolescents. Although reappraisal and acceptance are two of the most frequently used emotion regulation strategies that play an important role in health according to existing research results [46-48], adolescents may use various emotion regulation strategies in real life [46]. In the future, longitudinal behavioral and neuroimaging studies could be used to systematically compare the efficiency of implementing different emotion regulation strategies in optimistic or pessimistic adolescents, which will provide a key basis for interventions that could offer the most gain for subjective well-being of optimistic and pessimistic adolescents.

Reviewer 2 Report

Referee

Differential Effects of Optimism and Pessimism on Adolescents’ Psychological Well-Being: Mediating Roles of Reappraisal and Acceptance

The aim of this study was to explore how optimism and pessimism with a bi-dimensional structure differentially predicted satisfaction in life and depression in adolescence and the mediation role of reappraisal and acceptance in this prediction.

The paper introduces a very interesting topic. I think that methods and results are well carried out. My only concern is about the theoretical introduction. I think that Optimism and Pessimism need to be more specifically introduced. Also their specificity on adolescence and specifically on satisfaction in life and depression need to be more focused.  

Author Response

Response to Reviewer 2 Comments

Point 1: Differential Effects of Optimism and Pessimism on Adolescents’ Psychological Well-Being: Mediating Roles of Reappraisal and Acceptance

The aim of this study was to explore how optimism and pessimism with a bi-dimensional structure differentially predicted satisfaction in life and depression in adolescence and the mediation role of reappraisal and acceptance in this prediction.

The paper introduces a very interesting topic. I think that methods and results are well carried out. My only concern is about the theoretical introduction. I think that Optimism and Pessimism need to be more specifically introduced. Also their specificity on adolescence and specifically on satisfaction in life and depression need to be more focused.

Response 1: We would like to express our sincere thanks to you for the constructive and positive comments and suggestions.

According to your kind suggestions, we introduced optimism and pessimism more specifically from the perspective of their theoretical root. Based on the theoretical grounds of optimism and pessimism and their links to the approach and avoidance motivation systems, we detailed the possible specificity of optimism and pessimism specifically on adolescent life satisfaction and depression. The rewritten parts in the Introduction are marked in the revised manuscript.  

Reviewer 3 Report

The topic is of interest to both research and public policy. My main issues are:

1. Hypothesis 1 is far too vague, and even reads more like a research question. Please be more precise and specific, based on the theoretical framework.

2. The key concept "psychological wellbeing" is erroneous, it refers to meaning in life and eaudimonia. What the present paper targets is subjective wellbeing, which comprises life satisfaction and hedonia.

3. Cause-and-effect can not be established with certainty from a cross-sectional survey. Therefore, the authors need to be mindful of and clear about the fact that the findings are correlational relationships.

4. The text needs copy-editing, the flow is obstructed by typos and grammatical errors.

Best of luck with the continued work and thank you for an interesting read. 

Author Response

Response to Reviewer 3 Comments

Point 1: The topic is of interest to both research and public policy. My main issues are:

  1. Hypothesis 1 is far too vague, and even reads more like a research question. Please be more precise and specific, based on the theoretical framework.

Response 1: We would like to express our sincere thanks to you for the constructive and positive comments and suggestions. According to your kind suggestions, we added the theoretical root of optimism and pessimism, and detailed the possible specificity of optimism and pessimism specifically on adolescent life satisfaction and depression based on the theoretical grounds of optimism and pessimism and their links to the approach and avoidance motivation systems. This theoretical framework helps us to propose more precise and specific Hypothesis 1. The rewritten Hypothesis 1 is as follows:

Optimism would be a stronger predictor for adolescents’ life satisfaction than pessimism, and pessimism would be a stronger predictor for adolescents’ depression than optimism.

Point 2: The key concept "psychological wellbeing" is erroneous, it refers to meaning in life and eaudimonia. What the present paper targets is subjective wellbeing, which comprises life satisfaction and hedonia.

Response 2: Thanks for your constructive suggestions. Thanks to your suggestions and relevant literature, we learned more about the differences and similarities between subjective well-being and psychological well-being. Thanks a lot ! The concept “psychological well-being” has been replaced by “subjective well-being” in the revised manuscript.

Point 3: Cause-and-effect can not be established with certainty from a cross-sectional survey. Therefore, the authors need to be mindful of and clear about the fact that the findings are correlational relationships.

Response 3: Thanks for your kind suggestions. Yes, the cross-sectional design cannot support to establish cause-and-effect conclusion. We emphasized this limitation in the Discussion. The revised sentences are as follows:

The current study has several limitations that should be highlighted. First, the cross-sectional design of this exploratory study can provide valuable information for the research questions, but we should bear in mind that the findings are based on cor-relational data, which cannot support causal inferences. Future longitudinal or ex-perimental studies may be needed to clarify causal directions.

Point 4: The text needs copy-editing, the flow is obstructed by typos and grammatical errors.

Response 4: Thanks for your kind suggestions. According to your suggestions, we first checked the manuscript carefully. The manuscript then underwent English language editing by an experienced and native English speaking editor of MDPI. We hope that the text has been largely improved.

Point 5: Best of luck with the continued work and thank you for an interesting read. 

Response 5: Thanks again for all your comments and suggestions, which help us a lot to improve the manuscript.